# Supervised and Unsupervised Cell-Nuclei Detection in Immunohistology

Daniel Bug[1], Philipp Gräbel[1], Friedrich Feuerhake[2], Eva Oswald[3], Julia Schüler[3], and Dorit Merhof[1]

[1] Institute of Imaging and Computer Vision, RWTH Aachen University, Germany
`daniel.bug@lfb.rwth-aachen.de`
[2] Institute for Pathology, Hannover Medical School, Germany
[3] Charles River Discovery, Research Services Germany GmbH

**Abstract.** We introduce a simulation model for cell-nuclei in immunohistology that is enhanced by a cycle-consistent adversarial network to construct realistic virtual ground-truth data. This model is applied to the task of cell-nuclei detection in immunohistologically stained whole-slide images (WSI) and achieves a standalone median performance of 83.1% F1-score learning purely from synthetic annotations. We thoroughly evaluate different training scenarios with varying contributions of manual labels. It is shown that through the simulation model, the amount of required annotations can significantly be reduced without major performance losses. If only limited amounts of annotations are available, the simulation can lead to a stabilization in the detection of immune-cells.

**Keywords:** Immunohistology · Nuclei Detection · Virtual Ground-Truth

## 1 Motivation

Quantification of immunological markers is a frequent and important task in the analysis of digital microscopic images in histopathology. The detection of immune cell lineages or activation markers relies on specific diagnostic antibodies that are conjugated with chromogenic or fluorescent labels for microscopic analysis. With advances in digital pathology, cell counting can be automated to a large extent, supporting medical personnel by taking over time-consuming tasks such as large-scale quantification of microscopic image objects.

In recent research, numerous cell-nuclei detection and segmentation methods were presented [1, 4, 3, 6, 8, 9], with strong focus on hematoxylin and eosin (HE) stains. In the HE stain, cell-nuclei appear purple while the surrounding tissue is stained in pink. Immunohistochemically (IHC) stained images are different in appearance, since the antibodies used in this stain may bind the stain to a cell component other than the nucleus. In practice, analysis of specific binding to the cell-nucleus, cytoplasm or the cell membrane are relevant characteristics, see Figure 1 for an example from the dataset.

Algorithms built on destaining and blob-detection [1, 4] also perform well for cell-nuclei bound components, but fail to reliably detect cytoplasm and membrane bound stains. In this work, we address the challenging problem of IHC

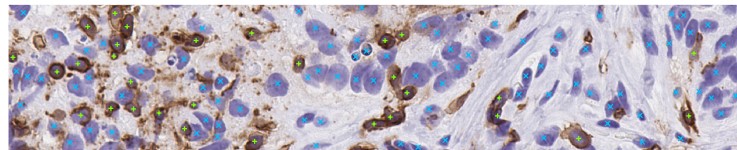

**Fig. 1.** Example of the binding characteristics in IHC stains. Ground-truth immune cells are denoted as green '+' and normal cell nuclei as blue 'x' (best viewed digital).

analysis of cytoplasm and membrane-binding stains by building a state-of-the-art deep learning pipeline and evaluating different training scenarios with a focus on the amount of annotations required. As the availability of labeled data often constitutes a limitation in medical image analysis, we aim at reducing the required amount of labels and incorporate synthetic ground-truth data from our developed simulation model. A manually defined basic simulation is improved by data driven enhancement using a Cycle-Consistent Generative-Adversarial Network (CycleGAN, [10]), which emerged as the state-of-the-art for unpaired image- and domain-translation. The lack of pairing is a double-edged sword: it enables training on nearly arbitrary image sets, but does not provide defined correspondence for the mapping. For example, [2] deployed a CycleGAN to translate different histological stains into a reference stain for a universal segmentation algorithm and added an $L_1$ loss to guide the initial phase of the training. In [5], a set of multiple CycleGANs has been employed to translate between a ground-truth domain (binary masks) and multiple organ domains for an organ-agnostic segmentation of nuclei in the HE stain. Contrary to our work, detection problems in HE lack the sub-classes of IHC stains that have to be reflected in the synthetic ground-truth generation, adding complexity to our task. Sub-classes enable various ways in which a CycleGAN can translate an image such that it appears valid for its domain, but does not reflect a corresponding ground-truth.

**Contribution:** We define a basic simulation model that can be enhanced by a CycleGAN for data augmentation in the training of a detection network. As unpaired image translations are not bound to form correspondences to the domain of origin, we formulate a transform-consistency loss to ensure a ground-truth correspondence. Furthermore, we evaluate different settings that consider training with purely synthetic ground-truth and compare this to scenarios that incorporate expert labels to varying extent.

In the title of this work, we refer to scenarios that exclusively rely on synthetic data as *unsupervised*, because no expert labels are used in training. *Supervised* then refers to scenarios that incorporate expert labels and thereby human supervision.

## 2 Method

A simulation model is a simple way to increase the amount of training data, thereby encouraging the network to learn certain useful data aspects. A brightfield-

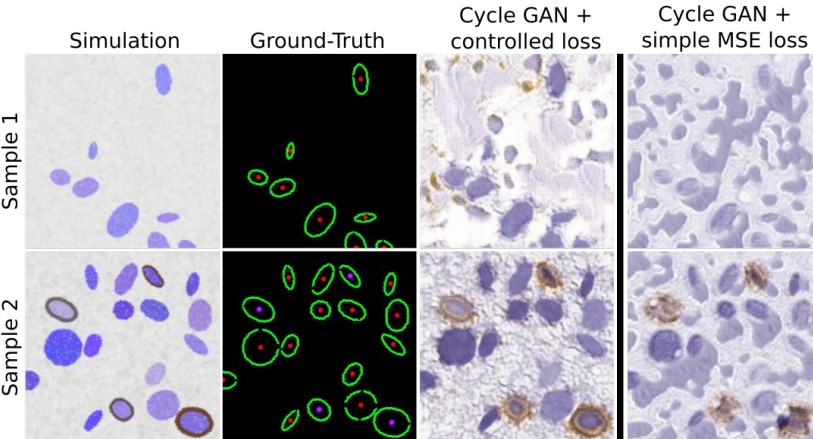

**Fig. 2.** From left to right: samples from the basic simulation, the synthetic ground-truth, the proposed CycleGAN-enhanced simulation with transform-consistency loss, and examples from a training with a standard MSE loss instead of the transform-consistency loss. The ground-truth shows the segmentation contour in green, all cell nuclei in red and immune cells in magenta.

microscopic image is constructed as follows: The background is simulated by iteratively adding shot-noise and applying a strong Gaussian blur to an average-white background. Between iterations, the image is normalized and clipped to ensure valid intensity ranges. We then place nuclei objects on this background at positions from a randomized grid using varying opacities to simulate different z-planes of the nuclei. A nucleus is modeled as circular object of blue color with a random radius sampled from a uniform distribution. The nucleus object is modified by random transforms to stretch and rotate it resulting in an arbitrarily oriented ellipse. To mimic inner nuclei structures, noise is added inside the ellipse. If an immune-cell is generated, a morphological gradient contour on the boundary is colorized in brown. Furthermore, we deploy the random elastic deformation recommended for U-net training [7].

A corresponding ground-truth image is generated by placing a small disk at the exact nuclei positions in the red channel (all cell-nuclei, CN) and additionally in the blue channel (immune cell-nuclei, IC). All CN boundaries are denoted as the morphological gradient of the simulated cell-nuclei. Pixels outside the CN are labeled as background. We refer to this as our *basic model*.

Since there is no notion of artifacts included and it represents only a very rough background model, the algorithm trained purely on the basic model operates in the high-recall – low-precision range.

To further improve the appearance with respect to the simulated data, we deploy a cycle-consistent generative-adversarial network [10] as a domain transfer mechanism between the simulation and real data. Samples from the simulation and its corresponding domain transfer are shown in Figure 2. For unpaired image

samples $S$ from the simulation domain and $R$ of the real WSI domain, the CycleGAN model consists of four networks: the generators $F : S \rightarrow R$ and $G : R \rightarrow S$ translate between domains, while the discriminators $D_S$ and $D_R$ estimate the probability of a sample belonging to their respective domain. The translation functions $F, G$ are realized by U-nets with four down- and upsampling stages, while the discriminators $D_S, D_R$ are built repeating the pattern: conv – instance norm – leaky ReLU repeated five times with downsampling after the first three blocks. From the definition in [10], we obtain two GAN losses $\mathcal{L}_{\text{GAN}}$ and the cycle-consistency loss $\mathcal{L}_{\text{cyc}}$ that contribute to the training:

$$\mathcal{L} = \mathcal{L}_{\text{GAN}}(F, D_R, S, R) + \mathcal{L}_{\text{GAN}}(G, D_S, S, R) + \lambda\mathcal{L}_{\text{cyc}}(F, G).$$

For the details on the different CycleGAN losses, please refer to [10].

While this loss enables a translation to images of similar appearance in the other domain, the image content may still change drastically as there is no loss term that explicitly defines a correspondence of structures. In consequence, the network is capable of changing spatial relations and to some degree 'imagining' new structures, which would be detrimental for an augmentation utilizing a synthetic ground-truth. This is not trivial to solve, as an additive pseudo-pairing by an MSE loss, see the last column in Figure 2, still results in major artifacts in the target domain. Thus, we contribute an additive transform-consistency loss $\mathcal{L}_T$ to enforce the image objects to remain in place and limit the changes to textural and color appearance:

$$\mathcal{L}_T = \alpha\text{MSE}_{\text{log}}(F(S), R) + \beta\text{MSE}_{\text{log}}(G(R), S),$$

where $\text{MSE}_{\text{log}}$ is the mean-squared-error computed from its logarithmic arguments and the hyperparameters $\alpha, \beta$ are loss weights. The logarithm is a design-choice to emphasize signals in brightfield microscopy, as cell-nuclei intensities are low and the background has high values. In our experience, $\beta$ can be chosen constant without restriction, while a fix choice of $\alpha$ introduces the dilemma of punishing a change we actually require, with little control about the consequences for gradient-descent learning. Herein, the challenge is to allow a certain amount of change for the translation, but just enough to ensure that cell-nuclei objects appear with consistent class and location (for example, to avoid splitting of cell-nuclei, random new objects, and object shadows). As a solution for more control over the learning behavior, we propose a feedback that intensifies the weight of this additional loss contribution dynamically and aims at maintaining a reference loss:

$$\alpha = \max(\alpha_{\text{ref}} + A(\mathcal{L}_T - \Phi_T), \alpha_{\text{min}}),$$

with a reference loss $\Phi_T$, reference contribution $\alpha_{\text{ref}}$, minimal loss contribution $\alpha_{\text{min}}$, and amplification $A$. In this formulation, the loss fluctuates around the reference contribution $\alpha_{\text{ref}}$, which typically is in the same order of magnitude as the other CycleGAN loss contributions. Note that $\mathcal{L}_T$ is now computed iteratively, first with $\alpha = 1$ then $\alpha$ is computed based on $\mathcal{L}_T|_{\alpha=1}$ and finally $\mathcal{L}_T$

is recomputed with the dynamic $\alpha$ value. $\alpha_{\min}$ ensures a positive transform-consistency loss contribution and should be a small positive number, in our case $\alpha_{\min} = 0.5$ (empirically chosen). $A$ was chosen empirically to increase the order of magnitude for unrealistic samples by a factor of 20. This choice was based on an initial training with small amplification, e.g $A = 1$, in which we monitored output examples during training to identify unrealistic outliers and artifacts. We observed a strong correspondence to $\mathcal{L}_T$ for these samples, which can conveniently be used to select a suitable range for $A$. From the same observation, $\Phi_T$ can be determined from output samples of decent quality.

Applying the U-net [7] for the actual detection, we predict the location of cell-nuclei centers, immune-cell-nuclei centers, the cell boundary and image background. Depending on the coupling properties of the stain-antibody, the predictions are not necessarily mutually exclusive. For example, in nuclei-bound components, the locations of a normal cell nucleus (CN) in hematoxylin blue and a highlighted immune-cell (IC) in diaminobenzidine brown are mutually exclusive. Membrane-bound components, however, characterize an immune-cell by a blue CN with brown surrounding. Thus, the detection can also be considered as multi-labeling task: CN positions with additional IC label, see magenta dots in Figure 2 and 5. This formulation implies using sigmoid output activations.

## 3  Evaluation

We extracted five regions-of-interest (ROI) from each of seven immunohisto-logically stained WSI and denoted nuclei positions and immune-cells. In total, the ROIs contain approx. 18000 annotated cell-nuclei (CN) of which 3000 are immune-cells (IC). On this dataset, we evaluate a slide-wise leave-one-out cross-validation (LOOCV) and the inverse one-versus-rest cross-validation (OVRCV) scenario to simulate a varying availability of expert annotations. As a second degree of freedom, we evaluate the contribution of simulated data to the performance, including the extreme cases of no real data and no simulated data. Throughout the training, the number of samples per epoch and the number of epochs are kept constant to guarantee that the respective scenario only influences the dataset's composition of real and synthetic data.

Performance is measured in terms of the F1-score, which requires definitions for a true positive (TP), false positive (FP) and false negative (FN). We define these as follows: TP means a prediction is within a 12px radius of a labeled nucleus, FP implies a prediction is outside this radius or multiple predictions are within the same radius, and FN means no prediction falls inside a labeled nucleus radius.

## 4  Results

Focusing on the extreme case of purely synthetic ground-truth for training first, we observe a strong increase in performance from the basic simulation to the CycleGAN enhanced model, as seen in Figure 3.

While the basic simulation model detects CN with a median F1-score of 71.3%, the domain transfer improves the detection to a median value of 83.1%. This is only a few percent lower than the best performance of the system using real data, see upper Figure 4. Although the F1-score for IC nuclei remains low, there is a notable 15% margin from the basic to the CycleGAN enhanced model.

In the LOOCV scenario without GAN augmentation (Figure 4, upper-left), we obtain an estimate for the performance using conventional annotation-driven training at a median value of 89.3% (CN) and 79.5% (IC). The median F1-scores decrease slightly to 88.7% (CN) and 77.6% (IC) when training with the CycleGAN augmentation, which is likely due to the influence of a small model error that is introduced by the simulation.

On the other hand, the OVRCV scenario models a system trained with a limited amount of annotations available (lower Figure 4). In this setting, the augmentation leads to an increase of the respective median F1-scores from 84.4% to 85.9% (CN) and from 63.9% to 67% (IC). Providing a realistic augmentation model enables at least a basic detection of IC, increasing the median from approx. 17% to 60% in slide two and from 52% to 69% in slide three. Additionally, it should be noted that with a slide containing numerous IC annotations, e.g. in slides one or seven, we obtain a performance in the overall CN detection that is close to the observed maximum, while training on only a fraction of the labels. In Figure 5, example images of the characteristics of the U-net detector is shown for different training conditions.

## 5 Discussion

Using a simulation in conjunction with an unsupervised domain transfer proved to be a viable model for training with virtual annotations in a cell-nuclei detection problem. Our technical contribution, a very strict domain transfer loss that controls the consistency between simulated and real-domain image, is necessary to utilize the virtual ground-truth generated by the simulation. This drastically reduces the workload in annotations and, in the extreme case of purely synthetic

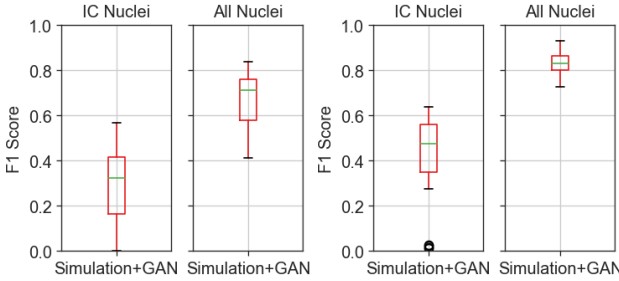

**Fig. 3.** Detection performance (F1-score) of the purely synthetic training scenario, based on the basic simulation (left) and a CycleGAN enhanced model (right). Evaluated on manually annotated real-data including all labels from all slides.

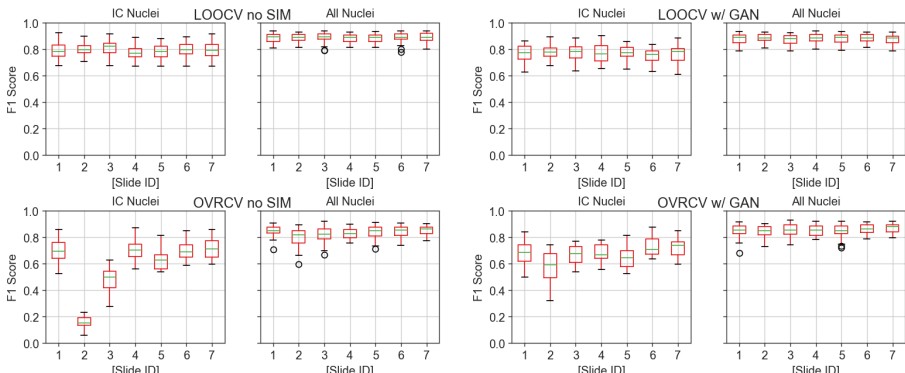

**Fig. 4.** Detection results under different training scenarios including real and simulated data. Upper row: leave-one-out cross-validation, lower row: one-versus-rest cross-validation, left: without simulated data, right: including the GAN-enhanced simulation.

training, entirely removes the requirement for expert supervision from the training. For medical image analysis, a data-driven training process with large labeled ground-truth is preferable, in practice. However, we see the strength of this approach not in an immediate application, but in a tool to quickly pre-generate labels for an expert supervision and semi-automatic annotation. To this end, the synthetic labeling approach facilitates rapid generation of an inital ground-truth to train a system that can continuously be enhanced. Notably, the majority of non-IC nuclei are detected reliably, while the more rare IC class likely requires manual correction. However, judging from the overall occurrence of CN vs. IC class, this already reduces the workload drastically, as there are many scenarios with far less IC than non-IC nuclei. For scenarios with high numbers of IC, the inclusion of representative labeled data to support the simulation model is to prefer. In cases of underrepresentation, looking particularly at slides two and three in lower Figure 4 (OVRCV scenario), the synthetic ground-truth can have a stabilizing effect. Furthermore, from the samples in Figure 5, we can assume that – despite the lack of quantitative measures – this model has the potential to learn a segmentation of nuclei boundaries implicitly from the simulation.

*Acknowledgements* This work was funded by the Federal Ministry of Education and Research, Germany (BMBF, FKZ: 031B0006A-C), as well as the German Research Foundation (DFG, Grant no. ME3737/3-1).

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
