# OpenReview forum: "Supervised and Unsupervised Cell-Nuclei Detection in Immunohistology"
_MICCAI.org/2019/Workshop/COMPAY — COMPAY 2019_

### Official Review · AnonReviewer3 · 2019-07-29
**The authors present a method for detection of nuclei in IHC images used a synthetic data generation approach.**

**Rating:** 6
**Confidence:** 5

**Review:**


The authors present a method for detection of nuclei in IHC images used a synthetic data generation approach.

Overall, I find the approach to be interesting and relatively novel. My major concern is that while the authors show their approach results in improvements from a baseline, the baselines itself is a bit contrived.

A more accurate baseline would have been against [5], which would show the precise benefit of the novelty taken in this approach in relation to the IHC component.

Of special note in that context is the statement “Notably, the majority of non-IC nuclei are detected reliably, while the more rare IC class likely requires manual correction”, while [5] is intended to produce the non-IC segmentations accurately, so the proposed novelty/value of this paper would seem to be entirely based on the results in the IC class. This would necessitate a deeper quantitative evaluation

Also, traditionally if making the claim that adding “some” manual annotations in proves things, one would expect to see a graph of performance versus number of manual annotations added to the training set to be able to examine the overall trend.

Authors should also be aware of this manuscript: "Optimized generation of high-resolution phantom images using cGAN: Application to quantification of Ki67 breast cancer images"
https://journals.plos.org/plosone/article?id=10.1371/journal.pone.01968461


Additional questions:
How long does training take?
How difficult was the training (is it a robust scheme or difficult like a GAN)?
What were the training parameters?
Can the authors speak to how well this would work for other primitives and use cases? Is this only destined to be used for IHC + nuclei segmentation?

---

### Official Review · AnonReviewer1 · 2019-08-14
**Interesting contribution to reducing the annotation workload in training cell-nuclei detection networks**

**Rating:** 7
**Confidence:** 3

**Review:**

This paper addresses the problem of how to drastically reduce the annotation workload in creating reference data for training neural networks to perform cell detection in immunohistochemically stained images. To this end a simulation model of cell nuclei is presented that is enhanced by a cycle-consistent generative adversarial network to obtain realistic virtual ground-truth data. Experiments on seven whole-slide images and different training scenarios with varying contributions of manual labels show that the use of the simulation model can indeed reduce the number of required annotations significantly without major performance losses.

Overall the paper is well written, makes an interesting contribution to the field, and presents promising results.

Specific comments:

- Section 3: "... we evaluate the contribution of simulated data ... including the extreme cases of ... no simulated data." Not clear what you mean.

- Section 3: "TP means a prediction is within a 12px radius of a labeled nucleus ..." What is the "12px" criterion based on? And how do the results of your experiments vary when varying this criterion?

- Section 4: "The median F1-scores decrease slightly ... when training with the CycleGAN augmentation, which is likely due to the influence of a small model error that is introduced by the simulation." That is a rather obvious statement. It would be more informative and more interesting to report on the origin and nature of the model error or how to find it.

---

### Decision · Program_Chairs · 2019-08-20

Accept